# A High-Content Microscopy Screening Identifies New Genes Involved in Cell Width Control in *Bacillus subtilis*

Dimitri Juillot,[a] Charlène Cornilleau,[a] Nathalie Deboosere,[b,c] Cyrille Billaudeau,[a] Parfait Evouna-Mengue,[d] Véronique Lejard,[d] Priscille Brodin,[b,c] Rut Carballido-López,[a] Arnaud Chastanet[a]

[a]Micalis Institute, INRAE, AgroParisTech, Université Paris-Saclay, Jouy-en-Josas, France
[b]University of Lille, CNRS, INSERM, CHU Lille, Institut Pasteur de Lille, CIIL-Center for Infection and Immunity of Lille, U1019-UMR 9017, Lille, France
[c]University of Lille, CNRS, INSERM, US41-UMS2014-PLBS, Lille, France
[d]Université Paris-Saclay, INRAE, MetaGenoPolis, Jouy-en-Josas, France

**ABSTRACT** How cells control their shape and size is a fundamental question of biology. In most bacteria, cell shape is imposed by the peptidoglycan (PG) polymeric meshwork that surrounds the cell. Thus, bacterial cell morphogenesis results from the coordinated action of the proteins assembling and degrading the PG shell. Remarkably, during steady-state growth, most bacteria maintain a defined shape along generations, suggesting that error-proof mechanisms tightly control the process. In the rod-shaped model for the Gram-positive bacterium *Bacillus subtilis*, the average cell length varies as a function of the growth rate, but the cell diameter remains constant throughout the cell cycle and across growth conditions. Here, in an attempt to shed light on the cellular circuits controlling bacterial cell width, we developed a screen to identify genetic determinants of cell width in *B. subtilis*. Using high-content screening (HCS) fluorescence microscopy and semiautomated measurement of single-cell dimensions, we screened a library of ~4,000 single knockout mutants. We identified 13 mutations significantly altering cell diameter, in genes that belong to several functional groups. In particular, our results indicate that metabolism plays a major role in cell width control in *B. subtilis*.

**IMPORTANCE** Bacterial shape is primarily dictated by the external cell wall, a vital structure that, as such, is the target of countless antibiotics. Our understanding of how bacteria synthesize and maintain this structure is therefore a cardinal question for both basic and applied research. Bacteria usually multiply from generation to generation while maintaining their progenies with rigorously identical shapes. This implies that the bacterial cells constantly monitor and maintain a set of parameters to ensure this perpetuation. Here, our study uses a large-scale microscopy approach to identify at the whole-genome level, in a model bacterium, the genes involved in the control of one of the most tightly controlled cellular parameters, the cell width.

**KEYWORDS** cell growth, cell shape, cell width, cell wall, HCS microscopy, *Bacillus subtilis*, Min system, Rod complex, cell division, metabolism, carbon metabolism

The bacterial landscape displays a rich variety of cell shapes, which are usually highly conserved at the single bacterial species level (1). The rationale behind a specific shape and its selective value remains speculative in most cases (1), as well as the molecular mechanisms that enable a specific shape to be determined and maintained across generations.

The shape of most bacterial cells directly depends on the shape of their cell wall (CW). The CW is primarily composed of a peptidoglycan (PG) scaffold that forms a rigid shell responsible for the mechanical properties of the cell envelope. In Gram-positive [G(+)] bacteria, the CW additionally contains PG-linked glycopolymers, the most abundant being the teichoic acids (TAs) (2). The PG sacculus is a contiguous matrix of linear sugar strands cross-linked by peptide bridges (3). Rod-shaped bacteria such as *Bacillus subtilis* and

Address correspondence to Rut Carballido-López, rut.carballido-lopez@inrae.fr, or Arnaud Chastanet, arnaud.chastanet@inrae.fr.

*Escherichia coli*, the models for G(+) and Gram-negative [G(−)] bacteria, respectively, use two different PG-synthesizing machineries, the divisome and the elongasome (4, 5). The divisome is required to build the septum at the site of division, which upon cell separation will become the new polar caps of the resulting daughter cells. The elongasome synthesizes the cylindrical sidewall during cell elongation. The latter comprises two machineries working semi-independently, one involving class A penicillin binding proteins (aPBPs), bifunctional enzymes with transpeptidase (TP) and transglycosylase (TG) activities, and one named the "Rod complex," which contains the RodA TG acting in concert with class B PBPs (bPBPs) carrying mono-functional TP activity such as PBP2A and PbpH in *B. subtilis* (5, 6). The prevailing model postulates that the Rod complex processively and directionally inserts glycan strands around the cell circumference, building the bulk of the PG meshwork, while aPBPs perform limited and localized, unoriented strand insertion (6–8). In agreement with this model, in *B. subtilis*, aPBPs are dispensable (9–11), while most PG synthases of the Rod complex are essential, such as RodA (6, 9, 12), or coessential, such as PBP2A and PbpH (13). This essentiality reflects that a failure in the proper establishment of the PG mesh compromises cellular integrity.

In addition to TP and TG enzymes, the Rod complex also includes the essential MreC and MreD morphogenetic proteins, which are presumed regulators of the activity of the complex (4, 14), and actin-like MreB proteins, which are believed to orient the circumferential motion of the complex (5, 15). The *B. subtilis* genome encodes three MreB paralogs, the essential MreB and Mbl, and MreBH, which becomes essential in the absence of the other two paralogs, in the absence of aPBPs, under stress conditions and at low $Mg^{2+}$ concentrations (11, 16, 17). RodZ, a protein of unknown function, is also a component of the Rod complex shown to be critical for rod shape maintenance in the G(−) bacteria *Caulobacter crescentus* and *Escherichia coli* and essential only in *C. crescentus* (18–21). The involvement of RodZ in shape control and its essentiality are less clearly established in G(+) bacteria. Described as essential in *B. subtilis* in an early report (22), several *rodZ* insertional or deletion mutants have been reported since, displaying minimal shape defects (23–25).

It has long been known that rod-shaped bacteria vary their size depending on the growth conditions and, in particular, on nutrient availability (26, 27). Rapidly growing cells have a bigger volume than slowly growing cells, a relationship often referred to as the (nutrient) "growth law" (for a review on this topic, see reference 2 or the very detailed reference 28). However, while in *E. coli*, cell width varies greatly (up to 100%) and concomitantly with cell length (26, 29–31), *B. subtilis* cells adjust their length but maintain a virtually constant diameter regardless of the growth conditions (31–35). This remarkable consistency suggests that cell width is a physiological parameter somehow encrypted in the genome of *B. subtilis* and that it must be carefully monitored during growth to correct for potential deviations to its nominal value. Yet how rod-shaped bacteria check and balance their diameter remains unclear. Recently, Garner and coworkers showed that the cell diameter results from the balance between the opposite activities of the Rod and aPBP elongation machineries (7). They proposed a model in which aPBP-mediated isotropic insertion of unoriented strands into the PG meshwork enlarges the cell cylinder while Rod complex-mediated organized circumferential insertion of PG strands reduces it (7). According to this model, the observation of thinner *B. subtilis* cells in the absence of aPBPs (36–38), can be explained as the result of the imbalance of the aPBP/Rod complex activities (7). Albeit thinner, cells that rely on the Rod complex for growth nevertheless retain their rod shape, indicating that the "check and balance" process of cell width control is still in place. Conversely, reduced activity of the Rod complex leads to the opposite imbalance, driving to an increased cell diameter (7). In the absence of the essential (or coessential) component(s) of the Rod complex, this ultimately leads to spherical cells, as exemplified by the depletion of RodA, MreC, MreD, PBP2A/PbpH, or MreB/Mbl/MreBH (13, 39–42). In agreement with this model, most genes reported to affect cell width in *B. subtilis* are directly involved in CW homeostasis, affecting one of the competing PG-synthesizing machineries, PG hydrolysis (required to allow PG expansion) or TA synthesis (Table S1). Other genes previously reported to affect width

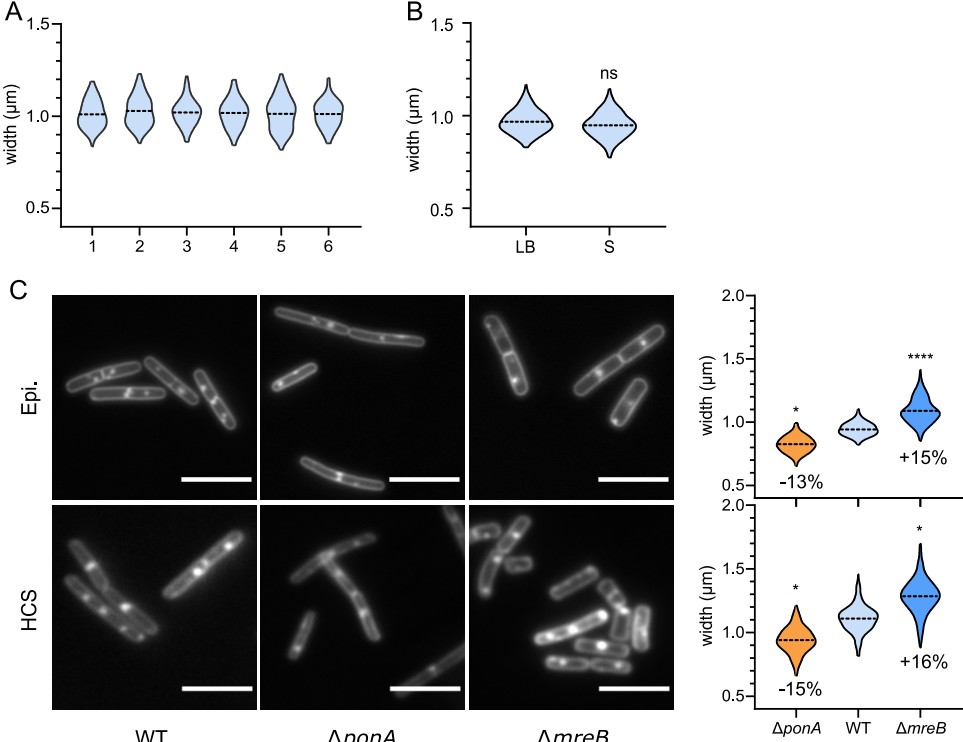

**FIG 1** Discrimination of diameter-control-deficient *B. subtilis* mutants: a proof of concept. (A) Comparison of cell width distribution of six independent cultures of fixed wild-type *B. subtilis* cells grown in rich (LB) medium, observed on an epifluorescence microscope. One-way ANOVA statistical analysis showed no significant differences between the replicates (Table S3). (B) Comparative cell width distributions of fixed wild-type *B. subtilis* cells grown in rich (LB) and minimal (S) media (epifluorescence microscope). (C) Qualitative (images) and quantitative (distribution of measured cell width) comparisons of data acquired on a wide-field epifluorescence microscope and a confocal HCSm, using the wild-type, Δ*mreB*, and Δ*ponA* mutant strains of *B. subtilis*. Fluorescent images were acquired on cells grown to the mid-exponential phase (0.2 < OD$_{600}$ nm < 0.3), fixed, and stained with FM1-43fx membrane dye. Discrete fluorescent foci result from cell fixation. Scale bar, 5 $\mu$m. Width distributions are displayed as violin plots with the broken line indicating the mean. Statistical analyses were performed as described in "Statistical Analysis." When significant, the difference between the means, expressed as a percentage, is indicated on the plots. Panels B and C are compilations of at least two independent experiments.

encode proteins whose absence perturbs the production or the localization of the latter (Table S1).

Here, we aimed at identifying at the genome-scale level additional determinants of cell width control during rapid exponential growth. We screened a complete organized collection of *B. subtilis* deletion mutants (24) using high-content screening microscopes (HCSm). Our protocol for midthroughput analysis allowed us to uncover several new genes that may work to maintain cell diameter. These are involved in several cellular processes, including CW synthesis, cell division, metabolism, and translation, suggesting that cell width homeostasis results from the combined action of several cellular circuits. Among these, our analysis suggests that metabolism and CW homeostasis are the two main routes affecting cell width.

## RESULTS

**Bacillus subtilis cells display limited width variability during rapid exponential growth.** It has long been accepted that, in contrast to *E. coli*, the cell diameter of *B. subtilis* cells remains virtually constant regardless of the growth rate (31, 32). We wondered how variable the cell diameter could be in isogenic *B. subtilis* populations during fast exponential growth. We used MicrobeJ, a Fiji plugin (43–45), to perform cell segmentation and quantify cell diameter (see Materials and Methods; Table S2). We first compared six independently acquired data sets of wild-type *B. subtilis* cells grown to the exponential phase in rich LB medium. We observed that the average width remained remarkably constant between

experiments (variability below 2%; Fig. 1A and Table S3). Also, the cell-to-cell variability (standard deviation) of the measured width in each population remained low, ranging from 0.071 to 0.089 $\mu$m across the different replicates (Table S3). These variations might reflect true differences of cell diameter or just the error of our measurements, but in either case variability was low. This reproducibility allowed us to expect the detection of potentially small variations of cell diameter between mutant strains. We next compared the diameter of *B. subtilis* cells exponentially growing in two different media, rich (LB) and poor (S), and thus supporting different growth rates. In agreement with previous reports (31, 32, 35), we found no significant difference of width between cells grown in rich and poor media (Fig. 1B). The cell-to-cell variability was similar in the two media, indicating that this variability is independent of the growth rate. Taken together, these experiments indicated that *B. subtilis* exerts a tight control over its diameter, whose variability remains below 2% on average across conditions and replicates.

**HCS microscopy allows screening for small phenotypic variations of cell width.** We next defined conditions that would minimize false positives in a microscopy-based screen of a genome-scale deletion library of *B. subtilis*. First, cells were fixed to obtain snapshots of their dimensions during exponential growth. Fixation induces a slight reduction of cell width relative to live cells (Fig. S1A) but prevents issues resulting from the time required for the preparation and imaging of multiwell plates with HCSm. Second, the growth medium was supplemented with 20 mM MgSO$_4$ to prevent potential inaccurate estimation of the cell diameter of mutants displaying irregular shapes or lysing. In *B. subtilis*, millimolar concentrations of magnesium in the growth medium are known to reduce the activity of PG hydrolases (46) and to alleviate the morphological defects of mutants affected in PG synthesis (47–49), allowing propagation of otherwise lethal mutations. Importantly, in the presence of high magnesium, these mutants display a normal rod shape but still present an abnormal width (47, 48). Addition of Mg$^{2+}$ to the growth medium slightly reduced the average width of wild-type cells (Fig. S1B), as previously reported (36). Our ability to detect these slight width differences when cells were either fixed (Fig. S1A) or grown in high magnesium (Fig. S1B) confirmed the sensitivity of our assay to detect small variations of average width between populations.

To further demonstrate the sensitivity of our assay, we tested the *mreB* and *ponA* null mutants, known to be wider and thinner, respectively, than wild-type cells (37, 50). As shown in Fig. 1C, the altered width of Δ*mreB* and Δ*ponA* mutants was unambiguously detected when cells were grown in high Mg$^{2+}$, fixed, and observed in either our conventional epifluorescence microscope or the HCSm. The cell-to-cell variability and the average cell widths noticeably increased when measurements were performed on HCSm-acquired images, but the relative difference of width between the two mutant strains and the wild type were perfectly conserved (Fig. 1C). These control experiments showed that mutants affected for the control of width could be identified in our medium-throughput HCS microscopy approach.

Next, we screened the complete *B. subtilis* kanamycin-marked ordered deletion library (BKK) (24), which contains 3,983 single-gene deletion mutants (~93% open reading frame coverage) of the parental 168 strain (GenBank accession number AI009126) (Fig. 2, see Materials and Methods for details). In order to prevent plate-to-plate fluctuations and to compare the widths of the mutants across plates, the width of each mutant was expressed relative to the average cell width per plate (AWP; see Materials and Methods). The average of the AWPs of the 48 plates (Fig. S1C) and the average cell width of the wild-type strain grown and imaged under the same conditions showed no significant difference (Fig. S1D). For each single mutant, we calculated the delta between its average width and the AWP of its plate (Table S4). The 3,983 Δwidth obtained displayed a Gaussian distribution, spreading from −13.9 to +23.4% but with 90% of the values contained in a narrow ±5% variation from the mean (Fig. 3A). Next, we arbitrarily set up a cutoff of the 1% most affected strains (0.5% largest and 0.5% thinnest) (Fig. 3A). The 40 mutants selected displayed a difference in diameter ranging from 8.9 to 23.4% of that of their AWP (Table S5, "Screening Step"). Using low-throughput epifluorescence microscopy imaging and the wild-type *B. subtilis* strain as a reference, we checked the cell width phenotype of the selected mutants (Fig. 2; see also Materials and Methods), while the deletion in each mutant was verified by PCR. Two of the

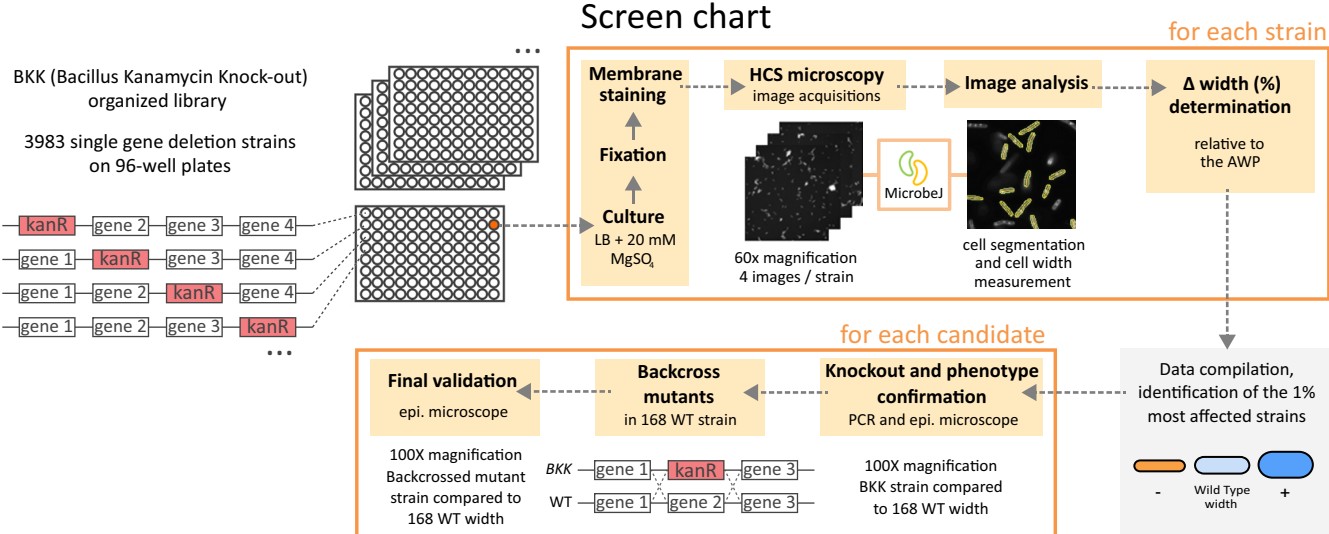

**FIG 2** Protocol summary of the screening process. Screening of the *B. subtilis* BKK collection arrayed in 96-well plates was based on automated image acquisition using an HCSm. The MicrobeJ plugin of Fiji was used for cell segmentation and width measurements. The average cell diameter of each mutant was compared to the average width of all cells on the plate (AWP). Candidates were confirmed by measuring their diameter on images acquired with an epifluorescence microscope, relative to the wild-type strain. The selected mutants were then backcrossed into a wild-type background before final width determination over triplicate experiments.

strains in the collection were wild type for the tested loci (Table S5) and were discarded for further analysis. A quarter of the mutants displayed a Δwidth of ≤2%, i.e., equivalent to the variability between wild-type replicates (Fig. 1A; Table S3), suggesting that our HCS microscopy analysis yielded some false positives (Table S5, "Checking Step"). All 38 confirmed knockout mutants were nevertheless backcrossed into the wild-type background (Fig. 2; Table S6) before attempting further characterization, in order to exclude phenotypes unlinked to the candidate gene deletions.

**Cell wall and central carbon metabolism genes are linked to cell width control.** Next, we carefully measured the diameter of the backcrossed mutants (Table S5, "Post-backcross Step"). A large reduction of the Δwidth compared to that of the parental strain was confirmed for most of them. Choosing a stringent Δwidth cutoff of 8%, we selected 12 mutants significantly and reproducibly (over 3 independent experiments) wider (*n* = 7) or thinner (*n* = 5) (Fig. 3B and C; Table 1; Table S5). We additionally kept the Δ*minJ* mutant despite its nonsignificant Δwidth (–1%) because of the peculiar uneven width affecting some of the cells in this mutant (Fig. 3C). The width phenotypes were conserved for all 13 mutants when grown without magnesium supplementation (Table S5, "Backcross Strains without Mg²⁺"), confirming that in contrast to cell bulging, swelling, and lysis (observed as consequences of CW synthesis impairment), magnesium cannot rescue the alteration of width. This suggests that cell width alteration does not result from uncontrolled PG hydrolytic activity.

Among the 13 selected mutants (Fig. 3B and C; Table 1), we identified 9 new genes affecting the cell width of *B. subtilis* (*ptsH*, *guaA*, *panD*, *ybzH*, *pyk*, *yaaA*, *minJ*, *dacA*, and *rpe*) and confirmed 4 others (*rodZ*, *cwlO*, *ftsE*, and *ftsX*) previously reported to be affected in cell diameter (Table S1; 'this study'). Note that several previously identified *B. subtilis* width deficient mutants are absent from the BKK due to their essentiality (e.g., *mreB*) (Table S1; "This Study"). Despite not being in the top 1% of genes retained for further analysis, the Δ*ponA* mutant still displayed a significantly reduced width in the first step of our screen (Table S4), as expected (Fig. 1C). However, the *mreBH*, *lytE*, and *rny* (*ymdA*) mutants did not display a significant width difference under our experimental conditions (Table S4). It should be noted too that a *rodZ* null mutant is present in the BKK library (24), even though the *rodZ* gene was originally reported to be essential in *B. subtilis* (22). We addressed this apparent discrepancy and showed that *rodZ* is not essential for growth in *B. subtilis*, at least under the experimental conditions tested. We also confirmed that Δ*rodZ* cells display division defects (51)

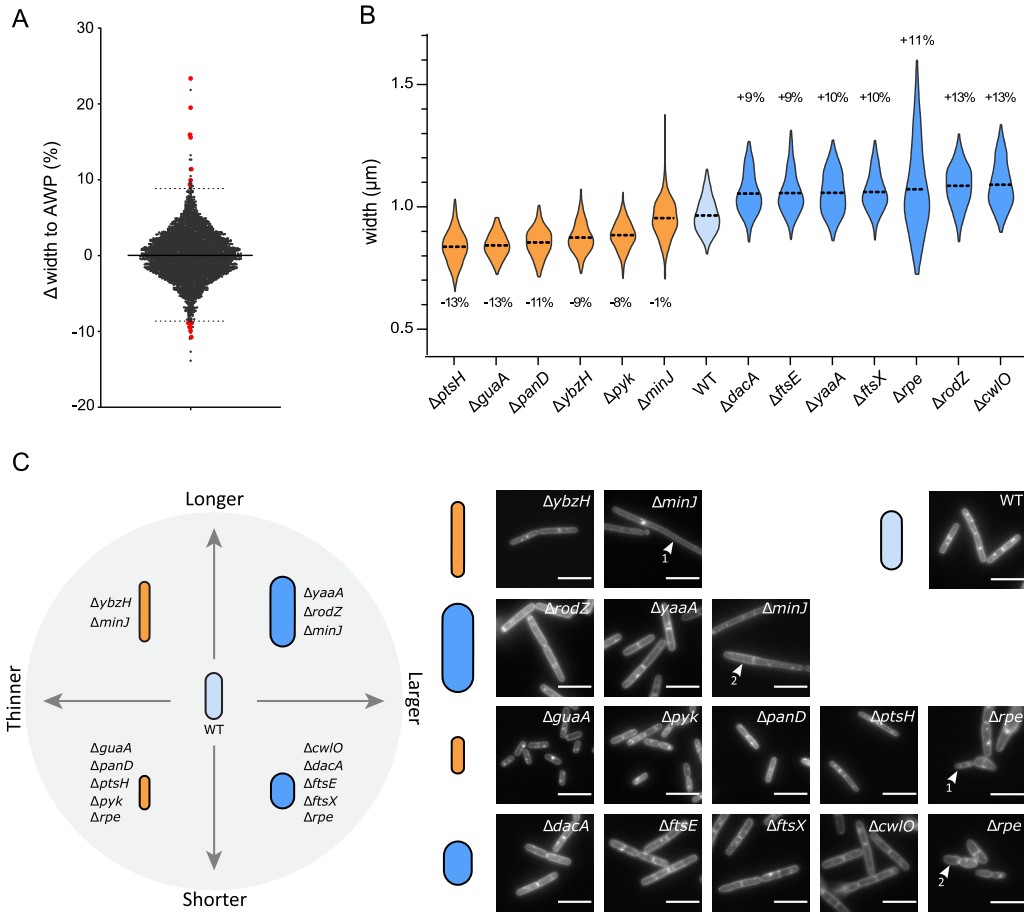

**FIG 3** The screen reveals 13 mutants with a cell width variation of >8% relative to the wild type. (A) Width difference (Δwidth) of each mutant relative to the AWP of its plate. Dotted lines indicate the cutoffs for the 0.5% largest (top) and thinnest (bottom) mutants. Red dots mark the 13 mutants with confirmed diameter defects after deletions were backcrossed into the wild-type strain. (B) Cell width distributions of the 13 selected (backcrossed) mutants. Orange and blue plots correspond to thinner and wider mutants, respectively, compared to their parental wild-type strain (light blue). Broken line, mean. Data are compilations of three independent experiments. The difference between the means of each mutant and the wild type is indicated, as a percentage. Statistical significances between the mutants and the wild-type strain width were calculated using nested *t* tests (see Table 1 for *P* values). All differences were significant except for the mean width of the Δ*minJ* mutant. (C) Phenotypes of the backcrossed mutants segregate into four classes based on their width and length defects. Δ*minJ* and Δ*rpe* mutants form both thinner (arrowhead, 1) and larger (arrowhead, 2) cells. Displayed are images of FM1-43fx membrane-labeled fixed cells. Scale bar, 5 μm.

and found that they display shape alterations in some media and that this phenotype is influenced by the parental genetic background (see Text S1 and Fig. S2).

The 9 new cell width determinants identified in our screen belong to different functional categories (Table 1). Interestingly, only one of them, *dacA* (encoding PBP5, a bPBP involved in PG maturation [52, 53]), is directly involved in CW homeostasis. The most represented functional category among our newly identified width-deficient mutants is metabolism (Table 1). One gene, *guaA*, is involved in purine nucleotide synthesis (encoding the GMP synthetase [54]), and four are part of the central carbon metabolism (Fig. S3A)—*pyk*, specifying the pyruvate kinase acting in glycolysis (55); *ptsH*, encoding HPr, a component of the sugar phosphotransferase system (PTS) (56); *panD*, involved in coenzyme A biosynthesis (57); and *rpe* (*yloR*), predicted to encode the ribulose-P-epimerase (Rpe) of the pentose phosphate pathway. Although the role of *rpe* has not yet been investigated in *B. subtilis*, the prediction got a score of >99.91% using a hidden Markov model-based homology prediction tool (HHpred; 58, 59). Of note, while most mutants involved in CW synthesis were wider, all metabolism mutants but *rpe* were thinner (Fig. 3B and C; Table 1). Out of the three remaining genes selected, one is involved in cell division (*minJ* [60, 61]), one is involved in translation (*yaaA*, encoding a ribosome assembly factor [62]), and one is annotated as a putative transcriptional regulator (*ybzH*

**TABLE 1** Cellular parameters of the confirmed width-control-deficient strains[a]

| Gene | Functional category[e] | δ length[b] (%) | P-value[c] | δ width[b] (%) | P-value[c] | δ GT[d] (min) | P-value[c] |
|---|---|---|---|---|---|---|---|
| *minJ* | cell division | 120.6 | **** | -1.0 | ns | 5,6 | * |
| *ybzH* | unknown (putative TR) | 12.9 | **** | -9.3 | **** | 1,7 | * |
| *rodZ* | CW homeostasis | 10.7 | ** | 12.5 | **** | -1,1 | ns |
| *yaaA* | translation | 4.4 | ** | 9.7 | *** | -0,7 | ns |
| WT | - | 0,0 | - | 0,0 | - | 0,0 | |
| *cwlO* | CW homeostasis | -5.2 | ns | 13.1 | *** | 0,3 | * |
| *ftsX* | CW homeostasis | -8.6 | ** | 10.0 | **** | -0,7 | ns |
| *ftsE* | CW homeostasis | -10.9 | **** | 9.4 | **** | -0,7 | ns |
| *dacA* | CW homeostasis | -11.7 | **** | 9.3 | **** | -1,1 | ns |
| *ptsH* | central C metabolism | -12.2 | **** | -13.3 | **** | 17,6 | * |
| *panD* | central C metabolism | -12.5 | **** | -11.4 | **** | 11,9 | * |
| *rpe* | central C metabolism | -13.6 | *** | 11.0 | **** | 12,8 | * |
| *pyk* | central C metabolism | -21.4 | **** | -8.3 | *** | 36,5 | * |
| *guaA* | purine metabolism | -36.7 | **** | -12.7 | **** | 38,1 | * |

[a]Colors of numbers indicate positively (purple) or negatively (orange) affected value in width or length, or increased or decreased generation time, compared to the wild-type strain.
[b]Difference (in %) relative to wild type, average of three independent pooled replicates.
[c]Statistical significance estimated by nested t-test (δ length, δ width) or Mann-Whitney test (GT). P-values are displayed as follows: **** = $P < 0.0001$; *** = $0.0001 < P < 0.001$; ** = $0.001 < P < 0.01$; * = $0.01 < P < 0.05$; ns = $P > 0.05$.
[d]Difference of generation time (GT) relative to the wild type, averages of four independent experiments.
[e]According to Subtiwiki (80).

[63]). Using the HHpred homology prediction tool (58, 59), we confirmed that *ybzH* encodes a probable helix-turn-helix (HTH)-type transcriptional regulator sharing strong structural resemblance with proteins of the ArsR and GntR families or transcriptional repressors. Regulators of the ArsR-type are involved in the stress-response to heavy-metal ions and GntR-family members in various metabolic pathways, including fatty acid, amino acid, and gluconate metabolism (64, 65).

**Δ*minJ* and Δ*rpe* display a phenotype of cell diameter instability.** The mutant strains selected in our screen displayed a thinner or a larger mean diameter relative to wild-type cells. Although their mean width differs from that of the wild type, most of these mutants still control their diameter to maintain it constantly over generations. However, two of these mutants, Δ*rpe* and Δ*minJ*, displayed a distinctive large dispersion of width values (Fig. 3B; Table S5).

The Δ*rpe* mutant from the BKK library (BKK15790) was first selected based on its reduced width (–11.4%) during the HCS microscopy analysis (Table S5). Surprisingly,

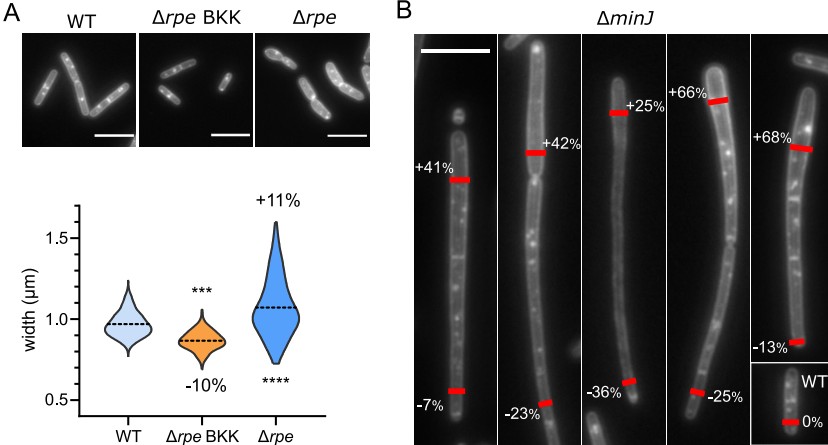

**FIG 4** Δ*minJ* and Δ*rpe* mutants exhibit an uncontrolled diameter phenotype. (A) Images of membrane-labeled strains and corresponding distribution of cell widths for the wild-type strain, Δ*rpe* from the BKK collection (Δ*rpe* BKK; BKK15790), and Δ*rpe* backcrossed into the 168 wild-type background (Δ*rpe*; RCL856). Broken line: mean. The differences between the means of the mutants and the wild type are indicated, as a percentage. Data are compilations of two independent experiments, and the statistical significance between mutants and the wild-type strain was calculated with a nested *t* test. (B) Images of membrane-labeled Δ*minJ* mutant (RCL834), presenting various widths along single cells or chains of cells. This phenotype affects a fraction of the population. Percentages indicate the difference of width compared to that of the wild-type strain at the position of the red marks. Scale bars, 5 μm.

mSystems®

once backcrossed, the Δ*rpe* strain (RCL856) displayed the opposite phenotype with an increased width (+11%) (Fig. 4A; Table S5), nonetheless indicative of a defect of width control. Another striking difference between the two strains was their width dispersion (Fig. 4A). While the BKK Δ*rpe* showed thin and regular cell diameters with a dispersion of values similar to that of the control strain, widths of the backcrossed mutant displayed the largest variability of our data set (Fig. 4A and 3B; Table S5). Furthermore, the backcrossed Δ*rpe* mutant formed small slow growing colonies, while its BKK parent did not (Fig. S3B). Taken together, these results suggest that the Δ*rpe* mutant present in our BKK collection had acquired some suppressor mutation(s), partially restoring its growth and reducing its width variability.

The second strain with a variable diameter was Δ*minJ* (Fig. 3B; Table S5). This mutant of the "Min" system, involved in division site selection, displays reduced septation, leading to long filamentous cells (Fig. 4B) (51, 60). Although the average width of Δ*minJ* cells was marginally affected (–1%), the SD was unusually large, with widths ranging from 0.7 to 1.4 $\mu$m (Fig. 3B and Fig. 4B). Furthermore, uneven diameters were observed along the length of individual Δ*minJ* filamentous cells. Single cells could display both wider and thinner widths than average, appearing tapered, although it is yet unclear if this dimorphism presents a polarity (Fig. 4B). This phenotype affected only a fraction of the population (up to 19%), which could explain why it was not previously reported.

**Mutants of metabolism and cell wall homeostasis are differently affected in growth and S/V ratio.** During the phenotypic characterization of the 13 width-deficient mutants selected in our screen, we noticed a variability of cell length too, with cells of the *guaA* mutant unambiguously being the shortest and cells of the *minJ* mutant forming very long cells (Fig. 3C). We quantified the average length of all mutants and found that, with the exception of *cwlO*, they were all significantly longer or shorter than the wild type (Fig. 5A, "Length"; Table 1). It should be noted that Δ*minJ* and Δ*rodZ* mutants form minicells (Fig. S3C) (22, 66) that were not taken into account in our length quantification. Interestingly, shorter and wider mutants were all related to CW homeostasis, while metabolism mutants were shorter and thinner than the wild type, again with the exception of Δ*rpe* (Fig. 3B and C and 5A; Table 1). However, no direct correlation between cell width and length was observed across the strains (Fig. S3D). Because cell length, but not width, of *B. subtilis* usually correlates with growth rate (the "growth law") (26, 67), we wondered if differences in cell length between the mutants would mirror differences in growth rate. The generation time (GT), determined during mid-exponential growth (see Materials and Methods), showed no significant difference with the GT of the wild type for CW homeostasis mutants (Fig. 5A, "GT"). However, the metabolism mutants displayed an increased GT of >63% relative to the wild type (Fig. 5A). For these strains, the GT strongly correlated with the average cell length ($R^2$ = 0.837), indicating that in such mutants the growth law is conserved (Fig. 5B). In contrast, no correlation was observed between their GT and their cell width (Fig. 5B), further indicating that these two parameters are not connected.

Finally, we calculated the surface area (S) to volume (V) ratio of the mutants. This parameter was proposed to be maintained constant in a given condition, as a key determinant of cell shape (68). The S/V was significantly altered in all our mutants, except Δ*minJ*. All CW mutants displayed a reduced S/V ratio (Fig. 5A, "S/V"), a consequence of the increased width and the subsequent increased cell volume (the length having a limited contribution to it [Fig. 5A, "length" and Fig. 5B]). This S/V reduction is reminiscent of the effect previously reported for fosfomycin-treated bacteria, an antibiotic inhibiting PG biosynthesis (68), and is consistent with the proposed model that reduction of the rate of S growth (i.e., when CW synthesis is reduced) increases cell width and reduces S/V (68). In contrast, all metabolism mutants but Δ*rpe* displayed a larger S/V as a consequence of the important drop of both width and length affecting the surface and the volume (Fig. 5A and B).

Taken together, our results discriminate between two main groups of width-deficient mutants with specific phenotypes. Mutants of metabolic genes (with the exception of Δ*rpe*) display a reduced width and increased S/V and are strongly impaired in growth, while conversely, mutants affected in CW homeostasis display an increased width and reduced S/V, but their GT is unaffected relative to the wild type.

**mSystems**®

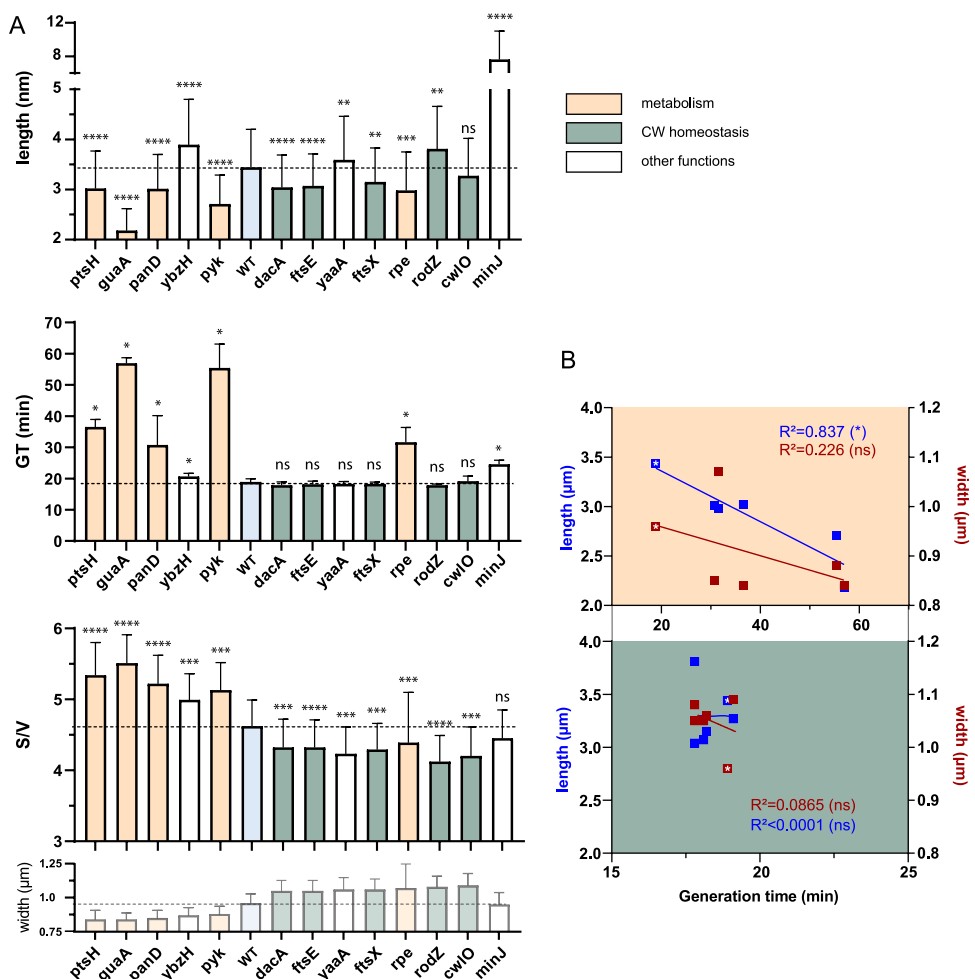

**FIG 5** Relationship between generation time, length, width, and surface to volume ratio in the selected mutants. (A) Average length, generation time (GT), and surface to volume ratio (S/V) of cell width-deficient (backcrossed) mutants compared to the wild-type strain. The width of each strain is recalled (from Fig. 3) for comparison. The dotted line marks the level of the average wild-type value. GT are calculated on populations (see Materials and Methods) and are the average of 4 independent experiments. Length and S/V ratio data are calculated per cell and compiled from three independent experiments. Statistical significance was determined as described in "statistical analysis" and displayed with * for *P* values. Error bars represent the standard deviation (SD). (B) Average length and width as a function of the generation time (GT). (Upper panel) Metabolism-related mutant; (lower panel) CW homeostasis-related mutant. R² of the linear regressions (lines) are indicated in the panels. White stars indicate the wild-type values.

## DISCUSSION

Cell width is probably one of the most tightly regulated physiological parameters in *B. subtilis* (34). However, the mechanisms allowing its fine control remain unclear. Our approach aimed at revealing in a systematic way nonessential genes involved in this process. We confirmed several of the previously reported nonessential genes acting on *B. subtilis* width control (Table S1) and identified 9 new genes whose deletion strongly affects *B. subtilis* diameter. Since we arbitrarily set up a cutoff to select the most drastically affected mutants (top 1%), it is likely that additional genes contribute to width control, along with essential genes (absent from the BKK library, such as *mreB*) or genes acting synthetically. A quick survey of our screen data with a less stringent cutoff (top 10% most affected mutants; Table S4) shows a few dozen genes involved in CW (e.g., *walH*, *pbpG*, *lytG*, *yocH*, *murQ*, *murE*, etc.), lipid metabolism (*fabI*, *lipL*, *araM*, *fadE*, etc.), and central carbon metabolism (*tkt*, *ywjH*, *coaA*, etc.). This list should nevertheless be taken with caution because the two-step verification performed on our top 1% selection revealed a significant number of false positives and

because the high-throughput-constructed BKK library may contain suppressors, as exemplified in this work with the *rpe* mutant.

To our knowledge, most genes previously described to affect cell width are directly involved in CW homeostasis (Table S1). The remaining genes (~1/4) are involved in a variety of pathways, but they were shown to affect the levels or localization of CW synthetic proteins or the levels of PG precursors (Table S1). In agreement with this, many mutants identified in our screen are related to CW homeostasis as well. In addition to *cwlO*, *rodZ*, *ftsX*, and *ftsE*, whose mutants were known to display width defects, we identified *dacA*, encoding the major vegetative DD-carboxypeptidase PBP5, responsible for the maturation of the PG by trimming the terminal D-Ala of the pentapeptide (69).

Unexpectedly, we also identified several genes involved in width control that belong to other functional categories, including five metabolic genes—*ptsH*, *guaA*, *rpe*, *pyk*, and *panD*. So far, the only metabolic gene described to affect cell width was *glmR*, which encodes a regulator controlling the carbon flux that stimulates the PG precursor synthetic pathway under neoglucogenic conditions (70, 71). Other studies have linked the cell metabolic status with cell size, although in these cases the mutants were affected in cell length (reviewed in reference 67). Out of these, only *pyk*, encoding PykA, which produces pyruvate in the final step of glycolysis (Fig. S3A), was identified in our screen, which may suggest a central role for this protein to coordinate the cell metabolic status with the control of length/division and width/elongation.

Another salient point of the study is that the two main groups of width-deficient mutants are discriminated by their phenotypes. Genes involved in CW homeostasis display an increased width and a reduced S/V but unaffected growth rate, while metabolism mutants display a reduced width and an increased S/V, and their growth rate is strongly affected. *rpe* stands out of this dichotomy by sharing characteristics of both groups, thus suggesting that its phenotypes might reflect a defect in both pathways. This hypothesis is strengthened by the presence of genes connected with cell shape control in the same operon as *rpe*—*prpC*, *prkC*, and *cpgA*. PrkC is a Ser/Thr kinase, and PrpC is its cognate Ser/Thr phosphatase, regulating many proteins, including some reported to affect cell width in *B. subtilis*, such as LtaS, YfnI, YqgS (72), CpgA (73), GlmR (YvcK) (74), RodZ (75), and GpsB (76) (Table S1). CpgA was recently shown to moonlight as a detoxifying enzyme of erythronate-4P, whose accumulation induces a depletion of fructose-6P, the entry of the PG precursor pathway (77). Thus, the *rpe* operon may be at the crossroad between the metabolic and CW homeostasis pathways.

Of note, the phenotypes of the CW mutants are consistent with the model proposed by Harris and Theriot (68). They proposed a "relative rate" model in which the rates of S and V growth are both functions of V (and not functions of S and V, respectively) and that S/V is the key parameter maintained constantly under a given condition rather than the respective rates of S and V expansion (68). A consequence of their model is that a diminishing rate of S growth, for example, when reducing the CW synthesis, both increases cell width and reduces S/V, even for a constant growth rate. Thus, one could hypothesize that the increase of width observed in the mutants identified in our screen may be a direct consequence of a crippled cell surface synthesis.

This could also be interpreted in light of the Rod/aPBP balance model from Dion and coworkers in which the unbalanced activity between the two CW synthetic machineries leads to thinner or larger cells (7). According to this model, the CW mutants selected in this study (*dacA*, *ftsE*, *ftsX*, *cwlO*, and *rodZ*) would present an unbalanced PG-synthesizing activity in favor of the aPBPs. Following the same line of thought, the metabolism mutants identified in this study, slender (with the exception of *rpe*), should present the opposite imbalance, with increased activity of the Rod complex or decreased activity of aPBPs.

In summary, cell width control appears as a very tightly regulated process in which different cellular circuits are at play. Our results indicate that metabolism is a major contributor to the control of cell width, suggesting the presence of unsuspected regulators or moonlighters affecting the synthesis of the CW. Among the genes identified here, 3 are stepping out and are of particular interest. On the one hand, RodZ acts on both cell division and elongation, and its activity depends on the medium composition (Fig. S2),

strengthening a possible link between metabolism and width. On the other hand, *minJ* and *rpe* mutant cells display unique uncontrolled width, suggesting that the check and balance of width control is lost. Deciphering how these genes affect the control of cell width of *B. subtilis* will be a challenge for future research.

## MATERIALS AND METHODS

**General methods and bacterial growth conditions.** Methods for growth of *B. subtilis*, transformation, selection of transformants, and so on have been described extensively elsewhere (78). DNA manipulations were carried out using standard methods. The *B. subtilis* strains used in this study are listed in Table S6. The *B. subtilis* strains were grown at 30°C or 37°C in rich lysogeny broth medium (LB), except for assaying growth in poor media, where strains were grown in modified salt medium (MSM) supplemented with 10 mM MgSO$_4$ (47) and S medium (32) with the corrected 1.2 $\mu$g/mL of MnSO$_4$. For precultures, medium supplements were added at the following final concentrations: MgSO$_4$, 20 mM; neomycin, 15 $\mu$g·mL$^{-1}$; spectinomycin, 100 $\mu$g·mL$^{-1}$; or chloramphenicol, 5 $\mu$g·mL$^{-1}$ (Table S6). Transformants were selected on LB agar plates supplemented with MgSO$_4$ and neomycin. For the determination of generation time (GT), cells from overnight cultures were diluted to a fixed starting optical density at 600 nm (OD$_{600nm}$) of 0.01 in fresh LB medium supplemented with MgSO$_4$ in 96-well cell culture plates (CellStar) and grown in a Synergy2 microplate reader (BioTek Instruments, Vermont, USA) at maximum rpm at 37°C. GT was calculated using a Matlab script available at https://github.com/CyrilleBillaudeau/GenerationTime_ofBacteria_withOD.

**General screening procedure.** Our screening was performed on the BKK library (24) using HCSm setups (see "High-Content Screening Microscopy," below), leading to the observation of 3,974 out of the 3,983 mutants from this collection (9 clones were absent from the published library or failed to regrow). Images were processed, and the cell diameter was measured (see "Image Processing and Cell Size Quantification," below). The 1% most affected strains (40 mutants) were selected and their phenotype confirmed using an epifluorescence microscope (see "Low-Throughput Epifluorescence Microscopy" and "Image Processing and Cell Size Quantification," below). Deletions in the selected clones were verified by PCR, revealing that 2 mutants (*yoqC*, *yorP*) were wild type for the expected locus, and thus they were discarded for further analysis. The remaining 38 mutants were backcrossed into the wild-type 168 strain to be analyzed over triplicate experiments using low-throughput microscopy. An arbitrary cutoff of 8% Δwidth, obtained by comparison to the wild-type strain width, was chosen, and 12 genes were finally selected.

**High-content screening microscopy.** Cells from overnight cultures, grown in the presence of neomycin and MgSO$_4$, were diluted at 1/600 in fresh LB medium supplemented with MgSO$_4$ in 96-well cell culture plates (CellStar) and grown on an orbital shaker at 250 rpm at 37°C until mid-exponential phase (OD$_{600}$, ~0.2). To fix the cells, 150 $\mu$L of culture was mixed with 50 $\mu$L of fixation solution (0.5 M KPO$_4$ pH 7, 8% paraformaldehyde, 0.08% glutaraldehyde) in 96-well PCR plates and incubated for 15 min at room temperature followed by 15 min on ice. The cells were pelleted by a 5-min centrifugation at 450 × *g*, and the supernatant was carefully removed by pipetting. The pellets were washed with 200 $\mu$L of washing buffer (KPO$_4$, 0.1 M, pH 7), centrifuged again, resuspended in 20 $\mu$L of water containing 3.3 $\mu$g/mL FM 1-43FX (Thermo Fisher; catalog number F35355) and incubated for 5 min at room temperature. Then, 180 $\mu$L of washing buffer was added, and the cells were centrifuged a last time to be concentrated 3.75× in 40 $\mu$L washing buffer. Next, 96-well (Fisher) or 384-well (Greiner) microscopy plates were treated with 60 $\mu$L of poly-L-lysine, 0.01%, and washed with 60 $\mu$L of deionized water; 40 $\mu$L of cells were put in each well and discarded after a 1-min incubation. Finally, 40 or 120 $\mu$L of deionized water was added into each well of 96-well or 384-well plates. Imaging was performed either on an ImageXpress micro confocal system (Molecular Devices) or an IN Cell 6000 analyzer (GE Healthcare) used in nonconfocal mode. The ImageXpress HCSm was equipped with a 60× Nikon air objective (numerical aperture [NA], 0.95), a fluorescein isothiocyanate (FITC) filter (Ex.488/Em.536), and a Zyla 4.2 Andor sCMOS camera with a final pixel size of 115 nm and controlled by the MetaXpress software package. The INCell 6000 analyzer was equipped with a 60 × water objective (NA, 0.95), an FITC filter (Ex.488/Em.525), and a sCMOS 5.5-Mpixel camera with a final pixel size of 108 nm and controlled by the INCell 6000 Analyzer Acquisition v.7.1 software. Images from 4 fields of view were acquired for each strain.

**Low-throughput epifluorescence microscopy.** Cultures were performed as for HCS microscopy but in shaking tubes instead of microplates. For live cell imaging, 300 $\mu$L of culture was directly mixed with FM 1-43FX (Thermo Fisher) to reach a concentration of 3.3 $\mu$g/mL and concentrated 3.75× before 1 $\mu$L of the preparation was spotted onto a thin 2% agarose-LB pad, topped by a coverslip and immersion oil, and mounted immediately in the temperature-controlled microscope stage. For the imaging of fixed cells, cells were fixed as described for HCS microscopy, except that 300 $\mu$L of culture was mixed with 100 $\mu$L of fixation solution and subsequently washed with 300 $\mu$L of buffer. Cells were spotted on a 2% agarose-LB pad or on a poly-L-lysine-treated 96-well microscopy plate. For the latter, the wells were washed and then filled with deionized water. Epifluorescence images of the membrane-stained cells were acquired on a previously described setup equipped with a 100 × objective (35).

**Image processing and cell size quantification.** The postacquisition treatment of the images was done with Fiji software, and the measurements (mean cell diameter and length) were done with the MicrobeJ plugin (43–45). In MicrobeJ, the cell width was calculated as the mean value along the medial axis of the cell. The parameters used for the MicrobeJ module are listed in Table S2. Cells aggregates were excluded, and segmentation was manually corrected when necessary. Note that the stained membranes appeared much larger and more pixelated on HCSm-acquired images than on the ones acquired on the epifluorescence microscope setup (Fig. 1C) due to the lower-resolution power of the HCSm (lower NAs) leading to a slight overestimation of the width by MicrobeJ.

During high-throughput screening, the cell width of each strain was calculated as the mean of 225 cells (on average). When the four-image set contained less than 30 measurable cells, a new acquisition was performed. The mean cell width of each mutant was compared to the AWP (<u>a</u>verage cell <u>w</u>idth <u>p</u>er plate) index, the average width of all measured cells of its 96-well plate (17 to $19.10^3$ cells/plate) (Fig. S1C). Each strain's diameter deviation relative to this index was calculated as

$$\Delta\text{width}\ (\%)\ =\ (\text{mean width of the mutant} - \text{AWP})\ \times\ 100/\text{AWP}$$

From these differences, the 0.5st and 99.5th percentile were calculated, and the 99% of the mutants between these two values was eliminated.

The use of an index resulted from the absence of a wild-type reference in each plate and in general in the BKK collection. Using a unique reference width (e.g., determined in Fig. 1A) for the calculation of the $\Delta$width is inappropriate because of small but unavoidable plate-to-plate variations (as seen in Fig. S2A) that would generate "plate biases." This could be prevented by using an internal reference per plate. Using the AWP index rather than introducing a wild-type-containing well on each plate presented the benefits of not relying on a single culture/well and of increasing the sampling (since it is calculated on 96 wells thus on $>10^4$ width values), thus producing a more robust reference. The AWP would be marginally affected by the presence of even a large number of strongly affected mutants on a single plate (e.g., 10 mutants with the highest average width observed in the present screen [+13%] on a single plate would affect their AWP by only 1%). The use of this index is possible because there is overall a low variability between strains, a limited number of strongly affected mutants (as seen in Fig. 3A), and we are selecting only the most highly affected mutants.

During low-throughput microscopy (for the verification of the BKK candidates and for clones resulting from the backcross into the 168 strain), the cell width and length of each strain were calculated as the mean of 245 cells (on average). The calculated $\Delta$width was expressed by comparison with the wild-type cell size.

**Alternative methods for cell width measurement.** Cell widths were measured either with the ChainTracer plugin of the Fiji software or by determining "manually" the width at the maximum height on intensity profiles (79) (Fig. S1A). For ChainTracer, we used a stack of phase-contrast and epifluorescence images of membrane-stained cells and only analyzed isolated chains of cells to prevent segmentation issues. For the measurement using intensity profiles, a line was manually drawn perpendicular to the cell's long axis on epifluorescence images of stained membranes, and a profile plot of the fluorescence intensity was generated. The cell diameter was determined by measuring the distance between the two maxima.

**Statistical analysis.** All statistical analyses were performed with Prism 9 (GraphPad Software, LLC). To analyze the variance between replicates, a multiple (pairwise) comparison was performed using one-way analysis of variance (ANOVA) (Fig. 1A). For pairwise comparison between means of a control and its tested sample with 2 or more replicates, we performed nested $t$ tests (Fig. 1B and C, 3B, and 4A; Fig. 5A, "Length" and "S/V"). Note that the plots show the pooled values of the replicates. When $t$ tests were not possible (e.g., if $n$ was $<30$), pairwise comparisons were done with a nonparametric Mann-Whitney test (Fig. 5A, "GT"). $P$ values are displayed as follows: ****, $P < 0.0001$; ***, $0.0001 < P < 0.001$; **, $0.001 < P < 0.01$; *, $0.01 < P < 0.05$; ns, $P > 0.05$.

## SUPPLEMENTAL MATERIAL

Supplemental material is available online only.

**TEXT S1**, PDF file, 0.3 MB.
**FIG S1**, PDF file, 0.1 MB.
**FIG S2**, PDF file, 0.4 MB.
**FIG S3**, PDF file, 1.7 MB.
**TABLE S1**, PDF file, 0.1 MB.
**TABLE S2**, PDF file, 0.05 MB.
**TABLE S3**, PDF file, 0.03 MB.
**TABLE S4**, PDF file, 0.2 MB.
**TABLE S5**, PDF file, 0.1 MB.
**TABLE S6**, PDF file, 0.1 MB.

## ACKNOWLEDGMENTS

This project has received funding (R.C.-L.) from the European Research Council (ERC) under the European Union's Seventh Framework Program (FP7) (no. 311231) and the Horizon 2020 research and innovation program (no. 772178).

We thank Alexandre Vandeputte and the BioImaging Center Lille-Nord de France (BICeL) facility (Lille, France) for HCS microscopy acquisitions.

Financial support for the HCS equipment was provided by the FEDER (12001407 [D-AL] Equipex Imaginex BioMed).

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
