## [Reviewer comments · mSystems]

A high content microscopy screening identifies new genes involved in cell width control in *Bacillus subtilis*

Dimitri Juillot, Charlène Cornilleau, Nathalie Deboosère, Cyrille Billaudeau, Parfait Evouna-Mengue, Veronique Lejard, Priscille Brodin, Rut Carballido-lopez, and Arnaud Chastanet

Corresponding Author(s): Arnaud Chastanet, INRAE

Review Timeline:

Submission Date:	August 10, 2021
Editorial Decision:	September 10, 2021
Revision Received:	October 8, 2021
Editorial Decision:	October 31, 2021
Revision Received:	November 3, 2021
Accepted:	November 4, 2021

Editor: Gilles van Wezel

Reviewer(s): Disclosure of reviewer identity is with reference to reviewer comments included in decision letter(s). The following individuals involved in review of your submission have agreed to reveal their identity: Dennis Claessen (Reviewer #3)

Transaction Report:

DOI: <https://doi.org/10.1128/mSystems.01017-21>

September 10, 2021

Dr. Arnaud Chastanet
INRAE
Jouy-en-Josas
France

Re: mSystems01017-21 (A high content microscopy screening identifies new genes involved in cell width control in *Bacillus subtilis*)

Dear Dr. Arnaud Chastanet:

Thank you for submitting your manuscript to mSystems. We have completed our review and I am pleased to inform you that, in principle, we expect to accept it for publication in mSystems. As you will see, both reviewers agree that this is very interesting work. At the same time, they raise a number of points that need to be addressed before the paper can be considered for publication.

Below you will find instructions from the mSystems editorial office and comments generated during the review.

Preparing Revision Guidelines

Sincerely,

Gilles van Wezel

Editor, mSystems

Journals Department
Reviewer comments:

Reviewer #1 (Comments for the Author):

This is a nice study describing a number of new cell width control mutants in *Bacillus subtilis*. The study setup described is elegant and robust as all the mutants identified are confirmed (or rejected) based on backcross experiments. Interestingly, next to genes involved in cell wall homeostasis, various metabolism genes are identified as important for width control. I have some comments on the (limits of) the methods that I would like to see addressed in the paper, and on the suggested link between observed phenotype and pathway affected. I also include a few suggestions regarding recently published (or preprinted) work that could be used to strengthen the discussion.

L 91 'a virtually constant diameter' - recent work from the Van Teeffelen lab, with very careful determinations of cell width, (<https://doi.org/10.1101/2021.05.05.442853>) revealed that cell width does vary at the individual cell level during steady state growth (although the range of variation is quite similar to what the authors describe in lines 128-9). The preprint also contains important notions on width control during the inhibition of various biosynthetic routes and it would be good if the authors reflect on the findings in this preprint in the discussion in this manuscript, also with respect to the discussion of the results in the light of the S/V model of Harris and coworkers.

ADP: First, why not use AWP - Average Width per Plate? The whole paper is about width control, and the 'ADP' is also derived from width measurements, but here the terminology suddenly changes to 'diameter'. I find this confusing. Second - there is quite some variation between measured ADPs in different plates - although I appreciate that the AVERAGE does not show a significant difference, an 'outlier' plate could result in the missing of width mutants. A case in point is the fact that the *ponA* mutant, used to benchmark the microscopy, is not picked up after the 1% cut-off (although its effect on width is stronger than that of the mutants that do make the cut-off). A reason for the plate-to-plate variation could be that, if indeed as shown in Fig. 2, the gene knockouts are sequential on a plate, the knocking out of several genes in one operon could result in a consistent (narrower/wider) phenotype, which would skew the ADP and thus result in false-negatives (as in missed) for identification of genes affecting width. Have the authors considered this? Did every plate contain a number of wildtype wells and was the width variation of wild type consistent with the variation in the entire plate? Also - *mreB* was not in the mutant collection. This should be mentioned explicitly.

249-251. Polarity of the uneven diameter phenotype of *minJ* mutants - it would be interesting to see if this can be linked to the 'age' of the pole - ie is it the pole generated by the most recent division that is always differently shaped or the other? This is relatively easy to do by a pulse chase growth experiment in the presence of clickable D-amino acid analogues to establish which part of the cell wall is the most recently generated. Also, *MinJ* localization both to the sidewall and poles/division sites has recently been determined using Superresolution Microscopy by the Bramkamp lab (<https://doi.org/10.1128/mBio.00296-21>).

L 319 Discrimination of width deficiency mutants by phenotype, with CW homeostasis mutants having increased width, and metabolism mutants decreased width (and prediction of functions of *yaaA* and *ybzH*). In my opinion, this is too strongly formulated. It is true that in this study, and with the cut-offs used by the authors, the CW homeostasis mutants resulted in increased width. It is however also true, and acknowledged by the authors (Table S1), that there are several CW homeostasis mutants that result in a decreased width. In fact, the authors speculate on the possible nature of the mutants in lines 341-347. Finally, have the authors considered the role of PG crosslinking on cell width? *PonA* mutant cells have a lower crosslinking degree and less monomers compared to wt, whereas *dacA* mutant cells have similar crosslinking, but less monomers and trimers suggesting that overall more MurNAc is engaged in 1:1 crosslinks (see Atrih et al 1999, PMC93885).

Table S1 - footnote '9' is used but not explained.
L 725 pAntothenate.

Reviewer #3 (Comments for the Author):

In this manuscript, the authors use a powerful microscopy screen to identify new genes involved in width control in the model *Bacillus subtilis*. The unbiased screen identifies known genes (assuring that the screen is working), and, interestingly, a series of new genes that were not known to be involved in width control. These genes mostly fall into two classes: those involved in metabolism and those involved in (regulation of) cell wall synthesis. The paper is well written and the work is of general interest. I only have a few small suggestions and questions

1. While LB medium is known to most (micro)biologists, the used S-medium is not so commonly used. Perhaps it's better to use "rich" medium and "poor" medium throughout.
2. Why are there small, but consistent differences in width when using the epifluorescence microscope and the HCSm setup (e.g. in Fig. 1C, where the average diameters are approx. 10% smaller when measured with HCSm)?
3. The surprising, and perhaps intriguing finding, is that the *rpe* mutant was detected as a mutant causing cells to be thinner, but after back-crossing indicates it's rather increasing the diameter, which the authors attribute to suppressor mutants. Such mutants could possibly be super interesting, as they may be involved in reducing width fluctuations. What is the evidence that there are suppressors in the original strain used to detect the aberrant width? I can see that the colonies are bigger compared to those obtained after the back-cross, but do the authors also detect suppressors in the back-crossed mutants?

4. In line with the previous comment. Is the BKK library in the same genetic background (168) as the authors used? Even between 168s there can be changes in morphology.
5. In Sup. Table 4 the words "average width" on top of each fourth column is not readable. It now reads "erage width (u"

Textual edits:

1. p.2, lines 44-47. I would rephrase this sentence. Also, replace the word "aim" by "target"
2. p.4, line 103: add "the" before "absence"
3. p. 5, line 113: Is HCSm High-Content Screening Microscopy rather than High-Content Screening Microscopes as mentioned now? This is also used in the M&M section.
4. p. 8, line 172: change "display" to "displayed"
5. p. 13, line 291: change "remains" to "remain"
6. p. 13, line 302: change "contains" to "contain"
7. p. 14, line 326: change "strengthen" to "strengthened"
8. p.14, line 339: change "hypothesizes" to "hypothesize"
9. p. 14, line 342: change "unbalance" to "unbalanced"
10. There are quite some inconsistencies in the reference list

Answers to reviewer comments:

We are thankful to both reviewers for their very nice comments and opinion on our work and their useful suggestions.

Reviewer #1 (Comments for the Author):

This is an nice study describing a number of new cell width control mutants in *Bacillus subtilis*. The study setup described is elegant and robust as all the mutants identified are confirmed (or rejected) based on backcross experiments. Interestingly, next to genes involved in cell wall homeostasis, various metabolism genes are identified as important for width control. I have some comments on the (limits of) the methods that I would like to see addressed in the paper, and on the suggested link between observed phenotype and pathway affected. I also include a few suggestions regarding recently published (or preprinted) work that could be used to strengthen the discussion.

L 91 'a virtually constant diameter' - recent work from the Van Teeffelen lab, with very careful determinations of cell width, (<https://doi.org/10.1101/2021.05.05.442853>) revealed that cell width does vary at the individual cell level during steady state growth (although the range of variation is quite similar to what the authors describe in lines 128-9). The preprint also contains important notions on width control during the inhibition of various biosynthetic routes and it would be good if the authors reflect on the findings in this preprint in the discussion in this manuscript, also with respect to the discussion of the results in the light of the S/V model of Harris and coworkers.

=> We are well aware of this manuscript as we have been in contact with their authors. We did not include it (among others) because we do not refer to un-reviewed articles as a general practice. In addition, although this manuscript present several very interesting observations about CW synthesis, and the relationship between cell mass, density, volume etc, we feel that it does not bring critical information relative to our question, at least in the current state of their manuscript. Regarding our focus, it is just confirming that the width parameter is invariable (from cell to cell and when varying growth conditions etc). Consequently, we prefer not to refer to this yet unpublished work.

ADP: First, why not use AWP - Average Width per Plate? The whole paper is about width control, and the 'ADP' is also derived from width measurements, but here the terminology suddenly changes to 'diameter'. I find this confusing.

=> This is a good suggestion. To be fair, we changed this acronym several time already, but if the current one is still confusing, let keep things simple. ADP has now been replaced with AWP throughout the text.

Second - there is quite some variation between measured ADPs in different plates - although I appreciate that the AVERAGE does not show a significant difference, an 'outlier' plate could result in the missing of width mutants.

=> It is in fact quite the opposite. If something is affecting an entire plate, using the ADP (AWP) allows to see the relative difference between a potential mutant of interest and the rest of the strains.

A case in point is the fact that the *ponA* mutant, used to benchmark the microscopy, is not picked up after the 1% cut-off (although its effect on width is stronger than that of the mutants that do make the cut-off).

=> The absence of *ponA* from the selection is not due to plate to plate variation. The width defect of *ponA* was simply lower during the screen than during the benchmarking. This is something that was expected: since the primary screen was too time-consuming to be replicated, mutants that would be “on a good day” during the screening, meaning that their delta-width (compare to wt) would be on their lower range, would not be selected because their width would be underestimated. The screen does not pretend to be exhaustive, and we indicated in the discussion that “it is likely that additional genes contribute to width control”

A reason for the plate-to-plate variation could be that, if indeed as shown in Fig. 2, the gene knockouts are sequential on a plate, the knocking out of several genes in one operon could result in a consistent (narrower/wider) phenotype, which would skew the ADP and thus result in false-negatives (as in missed) for identification of genes affecting width. Have the authors considered this?

=> The possibility raised by reviewer #1 is, we believe, unlikely. First, the genes are not always sequentially organized on the library (weirdly) and an operon could be split between different plates. Even if an operon was on a single plate and the width of each mutant was affected (+ or -), the overall effect on the entire plate would be negligible. For example, if we consider a case with a large 10-gene operon affected, with each mutant presenting a large +13% increased width (the maximum change observed here), this would rise the AWP by only 1%. It would need a lot more affected genes, or by a very large percent, to affect our ability to detect them. This approach works because there is overall a low variability between strains, a limited number of strongly affected mutants (as seen on Fig3A), and because we are using a stringent cutoff.

Also - *mreB* was not in the mutant collection. This should be mentioned explicitly.

=> We apologize for this. The information was present but hidden in the footnote of the Table Sup 1. This is now clearly stated in the result section (L.206-207) and in the discussion (L304).

(numbering refers to the “marked-up” version).

Did every plate contain a number of wildtype wells and was the width variation of wild type consistent with the variation in the entire plate?

=> As indicated in the M&M section (L441), there is no wild type on the BKK collection. That is why we used the AWP as a reference, which, as we explain hereafter, is *in fine* a better control. However we observed that the variation of width of the wild type was similar (in fact even greater) than the variation of width in the entire library (Fig. Sup 1D (former Fig S2)).

We did consider systematically replacing in some wells mutants by the wild type to use as a reference, but reasoned that the AWP was a better choice. First because the BKK library does not come with its parental wild type strain (we could not obtain it from the Gross lab despite our request) and using ours was taking a risk that its width may be slightly different (as raised by the 3rd referee). Second because each delta width would be calculated relatively to a single –or maybe couple –values for the wild type, that itself could fluctuate. If anything happened to the wild type

well, measurement on the entire plate would have been affected. The AWP is more robust since putative individual issues on one or the other well would not affect the average width and not jeopardize the measure. That was possible because 99% of the strains present a limited variation of width, and because we kept only the mutant with the greatest width defect.

249-251. Polarity of the uneven diameter phenotype of *minJ* mutants - it would be interesting to see if this can be linked to the 'age' of the pole - ie is it the pole generated by the most recent division that is always differently shaped or the other? This is relatively easy to do by a pulse chase growth experiment in the presence of clickable D-amino acid analogues to establish which part of the cell wall is the most recently generated. Also, MinJ localization both to the sidewall and poles/division sites has recently been determined using Superresolution Microscopy by the Bramkamp lab (<https://doi.org/10.1128/mBio.00296-21>).

=> the experiment suggested by reviewer1 does not require clickable analogs: a time-lapse allows to follow the history of the cell and to discriminate old/new poles (even though the clickable dyes offer other benefits). It seems that the phenotype of uneven width results from a two-step process of even widening followed by an uneven thinning of the cells. In other words, the two poles of the chain (large and thin) are "old". That said, this raises more questions than answers and it will require a thorough study to figure out how the absence of *minJ* affects the elongation and the same is true for each single mutant we have selected during this work. We respectfully don't think that adding this small piece of information would critically benefit the current manuscript and if agreed, we would prefer to keep it for a later study. Of note, we have noticed that the frequency of this phenotype was smaller than before. We are not sure what parameters influence this variation but, to be on the safe side, we have dampened our sentence (L258).

L 319 Discrimination of width deficiency mutants by phenotype, with CW homeostasis mutants having increased width, and metabolism mutants decreased width (and prediction of functions of *yaaA* and *ybzH*). In my opinion, this is too strongly formulated. It is true that in this study, and with the cut-offs used by the authors, the CW homeostasis mutants resulted in increased width. It is however also true, and acknowledged by the authors (Table S1), that there are several CW homeostasis mutants that result in a decreased width. In fact, the authors speculate on the possible nature of the mutants in lines 341-347.

=> When we discuss the point that CW mutants present an increased width (L328-331), we are not contradicting the literature and generalizing to every CW mutants, but only describing the phenotypes of the selected genes. However, we agree that our speculation about *yaaA* and *ybzH* was indeed crossing the line, and the corresponding sentence was therefore removed.

Finally, have the authors considered the role of PG crosslinking on cell width? PonA mutant cells have a lower crosslinking degree and less monomers compared to wt, whereas *dacA* mutant cells have similar crosslinking, but less monomers and trimers suggesting that overall more MurNac is engaged in 1:1 crosslinks (see Atrih et al 1999, PMC93885).

=> This is indeed an interesting observation and we are grateful to the reviewer to share his/her thought. This cannot explain the phenotypes of all the mutants identified (the crosslinking % seems

unaffected by the *cw/O* deletion for example (Sassine *et al.*, Sci. Rep. 2020)) but it may contribute to some of them indeed. We will certainly keep this in mind when digging into our mutants.

Table S1 - footnote '9' is used but not explained.

=> fixed

L 725 pAntothenate

=> fixed

Reviewer #3 (Comments for the Author):

In this manuscript, the authors use a powerful microscopy screen to identify new genes involved in width control in the model *Bacillus subtilis*. The unbiased screen identifies known genes (assuring that the screen is working), and, interestingly, a series of new genes that were not known to be involved in width control. These genes mostly fall into two classes: those involved in metabolism and those involved in (regulation of) cell wall synthesis. The paper is well written and the work is of general interest.

I only have a few small suggestions and questions

1. While LB medium is known to most (micro)biologists, the used S-medium is not so commonly used. Perhaps it's better to use "rich" medium and "poor" medium throughout.

=> In the main text, S medium is only mentioned in the first paragraph of the results as follows "We next compared the diameter of *B. subtilis* cells exponentially growing in two different media, rich (LB) and poor (S), (...)". We feel that it does not create too much confusion and is better for the reader to have the information that the poor medium is S without referring to the M&M, and let LB throughout since it is a fairly well-known medium. Thus we apologies but we kept it as it is.

2. Why are there small, but consistent differences in width when using the epifluorescence microscope and the HCSm setup (e.g. in Fig. 1C, where the average diameters are approx. 10% smaller when measured with HCSm)?

=> This difference originates from the combination of two factors: the different resolution power of the two kind of setup and the post processing analysis by ImageJ. Indeed, a notable difference between the setups is that the epifluorescence microscope has a higher resolution power due to the higher numerical aperture of its objective (theoretical diffraction limit in x/y ~190nm vs 250nm for the HCMs). It results from this that the stained-membranes appeared much larger and more pixelated on the HCSm-acquired images (visible on Fig1C). This in turn leads to a systematic overestimation of the width on these images. This is however not an issue since the relative width differences between the WT and the mutants (as exemplified in Fig 1) are conserved.

A sentence has been added to the M&M to explain this (L435-438).

3. The surprising, and perhaps intriguing finding, is that the *rpe* mutant was detected as a mutant causing cells to be thinner, but after back-crossing indicates it's rather increasing the diameter, which the authors attribute to suppressor mutants. Such mutants could possibly be super interesting, as they may be involved in reducing width fluctuations. What is the evidence that there are suppressors

in the original strain used to detect the aberrant width? I can see that the colonies are bigger compared to those obtained after the back-cross, but do the authors also detect suppressors in the back-crossed mutants? In line with the previous comment. Is the BKK library in the same genetic background (168) as the authors used? Even between 168s there can be changes in morphology.

=> The BKK library was constructed by the Gross lab in theoretically the same 168 strain than ours. We can't completely exclude the possibility of polymorphism between the WT strains, but we don't favor this hypothesis. The reason is that the backcrossed mutants forms small colonies and present a lytic phenotype on plate after a few days (but not the parent BKK mutant) from which colonies emerged. We streaked out a few outgrowing clones and some strongly resembled the BKK rpe mutant: they form large colonies without lytic phenotype and cells are thinner as well. This strongly suggests that suppressor mutations can be easily picked up, leading to the "BKK rpe" phenotype. We agree with the reviewer that this could be very interesting, and we hope to figure this out in the future. Since the description of this phenotype would bring limited information to the current manuscript, we have decided not to enter into such details.

5. In Sup. Table 4 the words "average width" on top of each fourth column is not readable. It now reads "erage width (u"

=> Fixed

Textual edits:

1. p.2, lines 44-47. I would rephrase this sentence. Also, replace the word "aim" by "target"

=> We have revamped most of the "importance" abstract and we are hoping it is now suitable.

2. p.4, line 103: add "the" before "absence"

=> Fixed

3. p. 5, line 113: Is HCSm High-Content Screening Microscopy rather than High-Content Screening Microscopes as mentioned now? This is also used in the M&M section.

=> our apologies for the inconsistencies. HCSm stands for "HCS microscope" and not 'microscopy', and we have now corrected the mistakenly used 'HCSm' by 'HCS microscopy' on L168, 187, 243, and 477.

4. p. 8, line 172: change "display" to "displayed"

=> Fixed

5. p. 13, line 291: change "remains" to "remain"

=> Fixed

6. p. 13, line 302: change "contains" to "contain"

=> Fixed

7. p. 14, line 326: change "strengthen" to "strengthened"

=> Fixed

8. p.14, line 339: change "hypothesizes" to "hypothesize"

=> Fixed

9. p. 14, line 342: change "unbalance" to "unbalanced"

=> Fixed

10. There are quite some inconsistencies in the reference list

=> indeed! Thanks for pointing this out. We have corrected many (hopefully all) mistakes in journal names and numbering.

October 14, 2021

Dr. Arnaud Chastanet
INRAE
Jouy-en-Josas
France

Re: mSystems01017-21R1 (A high content microscopy screening identifies new genes involved in cell width control in *Bacillus subtilis*)

Dear Dr. Arnaud Chastanet:

Thank you for submitting your manuscript to mSystems. We have completed our review and I am pleased to inform you that we will accept it for publication in mSystems. However, please address the comments by reviewer #1. Acceptance will not be final until you have adequately addressed the reviewer comments.

I agree with the reviewer on the subject of minJ, so I would suggest taking those suggestions on board. Please also see if you can expand the methodology a bit more. As for preprints, my view would be that if online preprints come out that shed new light on conclusions in the paper, that should be incorporated. I however feel that for your paper, such is not the case, and I therefore leave that decision up to your team.

Preparing Revision Guidelines

Sincerely,

Gilles van Wezel

Editor, mSystems

Journals Department
Reviewer comments:

Reviewer #1 (Comments for the Author):

The manuscript has been greatly improved and I thank the authors for their careful response to the points raised. I have a few comments:

- I raised a few points regarding methodology that to me were not clear from reading the MS. These points are explained in the response but it may be helpful for people reading the paper if the methodology was explained a bit more extensively in the main text, especially where it concerns controls.
- I find it unfortunate that the authors stick to an old-fashioned view of the scientific literature where a paper only becomes valid once it has been checked by an arbitrary number of colleagues who perform peer review. We all know peer review has its flaws, and preprints are a great way of sharing scientific findings before peer review. Of course, one does not need to interpret everything in a preprint as 'valid', and can perform one's own version of review on a preprint, but that does not mean that one should not use the knowledge present in preprints.
- I don't agree with the answer on the minJ mutant that both poles in a filament are old. Although a filament represents a number of missed division events, one of the poles in the filament will always be older than the other. If the authors do not wish to dig into this deeper, the remark about the 'polarity' of the phenotype should be altered, and these shapes should be described as tapered with an explicit remark that it is unclear whether there is any polarity to this tapering.

Reviewer #3 (Comments for the Author):

I would like to thank the authors for carefully revising their manuscript. I have no further comments.

Answers to reviewer comments:

We are thankful to the reviewer for his/her precious remarks and to help us improving this manuscript.

Reviewer #1 (Comments for the Author):

The manuscript has been greatly improved and I thank the authors for their careful response to the points raised. I have a few comments:

- I raised a few points regarding methodology that to me were not clear from reading the MS. These points are explained in the response but it may be helpful for people reading the paper if the methodology was explained a bit more extensively in the main text, especially where it concerns controls.

=> We tried our best to clarify our methodology in particular concerning the absence of a wild type control on the library and the benefits and robustness of the index (awp). A paragraph has been added (L442-453) in the method section. We hope this is what the reviewer had in mind.

- I find it unfortunate that the authors stick to an old-fashioned view of the scientific literature where a paper only becomes valid once it has been checked by an arbitrary number of colleagues who perform peer review. We all know peer review has its flaws, and preprints are a great way of sharing scientific findings before peer review. Of course, one does not need to interpret everything in a preprint as 'valid', and can perform one's own version of review on a preprint, but that does not mean that one should not use the knowledge present in preprints.

=> We understand the reviewer view on the use of preprints. We agree that the peer-review system has its flaws, but preprints are not devoid of such issues either. And -for the sake of arguing- if unreviewed works can be cited, used for CV and give all the benefits of reviewed works, what would be the benefits of publishing papers at all? Now we agree that a draft paper that we, as a peer, would consider solid and that could bring important information, may be mentioned, but it does not seem to be the case here.

- I don't agree with the answer on the minJ mutant that both poles in a filament are old. Although a filament represents a number of missed division events, one of the poles in the filament will always be older than the other. If the authors do not wish to dig into this deeper, the remark about the 'polarity' of the phenotype should be altered, and these shapes should be described as tapered with an explicit remark that it is unclear whether there is any polarity to this tapering.

=> We are sorry about our unclear answer. Yes, we agree that one pole is necessarily older than the other. We have amended the text according to the reviewer suggestion (L253-5).

Reviewer #3 (Comments for the Author):

I would like to thank the authors for carefully revising their manuscript. I have no further comments.

=> thank you

November 4, 2021

Dr. Arnaud Chastanet
INRAE
Jouy-en-Josas
France

Re: mSystems01017-21R2 (A high content microscopy screening identifies new genes involved in cell width control in *Bacillus subtilis*)

Dear Dr. Arnaud Chastanet:

Your manuscript has been accepted, and I am forwarding it to the ASM Journals Department for publication. For your reference, ASM Journals' address is given below. Before it can be scheduled for publication, your manuscript will be checked by the mSystems senior production editor, Ellie Ghatineh, to make sure that all elements meet the technical requirements for publication. She will contact you if anything needs to be revised before copyediting and production can begin. Otherwise, you will be notified when your proofs are ready to be viewed.

As an open-access publication, mSystems receives no financial support from paid subscriptions and depends on authors' prompt payment of publication fees as soon as their articles are accepted. =

Publication Fees:

We recognize that the video files can become quite large, and so to avoid quality loss ASM suggests sending the video file via <https://www.wetransfer.com/>. When you have a final version of the video and the still ready to share, please send it to Ellie Ghatineh at eghatineh@asmusa.org.

Sincerely,

Gilles van Wezel
Editor, mSystems

Journals Department
Table S6: Accept
Supplemental Figure 3: Accept
Supplemental Figure 1: Accept
Table S4: Accept
Table S3: Accept
Table S5: Accept
Table S1: Accept
Table S2: Accept
Supplemental Figure 2: Accept
Supplemental information: Accept